# Update on the Management of Uveitic Macular Edema

**DOI:** 10.3390/jcm10184133

**Published:** 2021-09-14

**Authors:** Slawomir Jan Teper

**Affiliations:** Clinical Department of Ophthalmology, Faculty of Medical Sciences in Zabrze, Medical University of Silesia, 40-760 Katowice, Poland; slawomir.teper@sum.edu.pl

**Keywords:** macular edema, uveitis, uveitic complications, non-steroidal anti-inflammatory drugs, steroids, biologic treatment

## Abstract

Uveitic macular edema (ME) is a frequent complication in 8.3% of uveitis patients and is a leading cause of serious visual impairment in about 40% of cases. Despite the numerous available drugs for its treatment, at least a third of patients fail to achieve satisfactory improvement in visual acuity. First-line drugs are steroids administered by various routes, but drug intolerance or ineffectiveness occur frequently, requiring the addition of other groups of therapeutic drugs. Immunomodulatory and biological drugs can have positive effects on inflammation and often on the accompanying ME, but most uveitic randomized clinical trials to date have not aimed to reduce ME; hence, there is no clear scientific evidence of their effectiveness in this regard. Before starting therapy to reduce general or local immunity, infectious causes of inflammation should be ruled out. This paper discusses local and systemic drugs, including steroids, biological drugs, immunomodulators, VEGF inhibitors, and anti-infection medication.

## 1. Introduction

Uveitis is a common cause of blindness, especially at working age and in low- or middle-income countries [1,2]. There are many causes of blindness or significant visual impairment in patients with uveitis, but the most common and important is macular edema (ME), which affects about 40% of patients with 20/60 visual acuity or less, according to [3]. There are many causes of ME, but secondary edema caused by uveitis has perhaps one of the most complex and varied pathomechanisms [4]. ME occurs in 8.3% of non-infectious uveitis patients [4]. It can persist without any sign of concurrent inflammation, but active inflammation can make ME difficult to treat.

Inflammatory ME is a complication of a heterogeneous group of diseases with complex etiologies, which makes multicenter clinical trials for uveitic ME difficult and expensive, and a lack of high-quality evidence-based medical data hinders guideline development. Despite the striking increase in related studies in recent decades, many clinical decisions in uveitis cases are not supported by strong scientific evidence. This paper therefore reviews the available methods of treating ME, together with their advantages and disadvantages, and indicates appropriate therapies for specific clinical situations. Proper uveitis management that considers the etiology in a specific case is sometimes enough to restore normal retinal thickness, but many patients require additional treatment dedicated to ME. Infectious cases, in which an eradication of the infection suffices, are an exception. Thus, this paper proposes treatment for ME following infections, focusing on the treatment of ME itself rather than etiological factors. Acute treatment usually involves the use of steroids via various routes of administration. Long-term treatment, however, should avoid the use of steroid drugs because of their common side effects.

## 2. Infectious Uveitis

### Anti-Infection Agents

ME may be secondary to an infection, and aggressive steroid or intravitreal treatment should not be started until infection has been ruled out. Table 1 lists the most common pathogens involved in ME, together with examples of treatment regimens. Besides the pathogens listed in the table, many other viruses, bacteria, fungi, and parasites can be responsible for ME.

## 3. Non-Infectious Uveitis

### 3.1. Local Treatment

The advantage of local eye treatment is usually its negligible effect on other organs. The disadvantage is that it treats only one eye, whereas inflammatory ME often affects both eyes. Moreover, obtaining a high drug concentration requires either a frequent use of the drug or an invasive administration route. The emergence of long-acting drugs administered locally has contributed to the increasing popularity of such treatment.

#### 3.1.1. Topical Steroids

Corticosteroids are first-line agents for addressing inflammation in most acute uveitis cases. However, the topical route of administration limits their effectiveness to only mild ME following anterior uveitis. Dexamethasone can be used frequently. Usually, when initiating treatment for acute anterior uveitis, corticosteroids are given every 1 h except during the night. Although application every 15 min can be even more effective, this dosing is only possible for a brief period. As the inflammation subsides over the following days or weeks, the eye drops can be used less frequently, but discontinuation of treatment before at least a few weeks may cause a rapid relapse. In terms of ME, it can be used only in cases of anterior uveitis—usually for patients with rather mild or no ME. No reports have confirmed their efficacy for treating uveitic ME.

#### 3.1.2. Topical Non-Steroidal Anti-Inflammatory Drugs (NSAIDs)

The activity of cyclooxygenases during inflammation promotes the increased production of prostaglandins, which in turn increases vascular permeability and contributes to ME. Although NSAIDs seem to be an excellent treatment choice, pseudophakic ME does not share the same cytokine profile as uveitic ME, and clinical trials have not confirmed the efficacy of topical drugs; hence, they are now used for pseudophakic ME or as a controversial adjunct to corticosteroids for uveitis. Their effectiveness for uveitic ME has not been clearly proven, and their effects are often of borderline statistical significance [12]. Scientific publications have mentioned, for example, bromfenac and nepafenac for this indication [13,14]. Usually, NSAIDs are administered three times daily, but bromfenac seems to be sufficient twice a day for pseudophakic ME [13].

#### 3.1.3. Topical Interferons

Topical interferons are used off-label because there is no topical formulation on the market, but they should be prepared by a pharmacy. In many countries, the availability of INF-α2a (Roferon-A^®^) or INF-α2b (Intron-A^®^) is limited, and scientific reports on their topical administration are also limited; consequently, they are rarely prescribed. INF-α and TNF-α have opposite effects in inflammation. Reports have shown the efficacy of topical administration four times a day for both uveitic and diabetic ME [15,16,17].

#### 3.1.4. Periocular Steroids

Periocular drug administration can be divided into subconjunctival, peribulbar, and retrobulbar injections; injections under the Tenon capsule; and a relatively new route, suprachoroidal injections. In each of these cases, the drugs used for uveitic ME are steroids.

##### Subconjunctival Steroids

For inflammation in the anterior region of the eye, subconjunctival steroids are often used, but their use for ME is much less frequent and is associated with lower penetration into the posterior region and a relatively short duration of action, especially for dexamethasone.

Frequent (e.g., five times a day) subconjunctival dexamethasone injections increase steroid concentration in the eye, but are inconvenient for the patient and the ophthalmologist, and the efficacy is transitory. Since treatment of ME is necessarily long term, dexamethasone administration is not used for this purpose.

The use of long-acting steroids seems to be the right choice for subconjunctival administration. However, although further studies have been called for, especially for economic reasons, no multicenter randomized studies have compared this option with others [18].

So far, the published results for such treatment are promising—CST reduction occurs in most patients, with accompanying improvement in visual acuity and relapses in only about a quarter of patients six months after an injection [19]. Unfortunately, this route of administration can have side effects, especially in the form of an increase in IOP (25% of participants), with some patients requiring surgical rinsing out of triamcinolone.

Other reported local side effects, such as conjunctival ulceration, necrosis, and infectious scleritis, are rare [20,21].

##### Subtenon/Peribulbar Steroids

Injections under the Tenon capsule have been performed for over 50 years, and although the first report concerned optic neuritis [22], it was quickly realized that this route was also useful for uveitis [23]. A complication for ophthalmologists performing this procedure is possible perforation of the eyeball, which can be avoided by moving the needle sideways during the injection.

As with the administration of steroids by other routes, an increase in IOP is a common adverse effect depending on the dose and location of the drug [24]. Rearward injections appear to have a lower risk of significantly increasing IOP [25].

Elevated IOP (>21 mm Hg) can be expected in about 15–20% of uveitic patients following triamcinolone injections, and an increase in IOP above 5 mm Hg has been observed in roughly a third of patients [26]. However, some papers have reported ocular hypertension in more than 75% of patients receiving subtenon triamcinolone, with a 23% incidence of glaucoma in the follow-up period [27].

##### Suprachoroidal Route

The method of delivering medications between the choroid and the sclera was devised to increase the concentration of drugs in the posterior pole of the eye more effectively than administration under the Tenon capsule or into the vitreous. This method is still under investigation, but the results so far are very promising, with a special triamcinolone formula administered in this way being developed. The results of the six-month phase 3 PEACHTREE study showed that the efficacy and safety profile was satisfactory, and this was confirmed by the MAGNOLIA study. Almost half the patients who were treated with suprachoroidally injected triamcinolone acetonide formulation (CLS-TA)—a suspension of triamcinolone acetonide—gained 15 or more ETDRS letters for BCVA versus 16% in the control group (*p* < 0.001), and the mean improvement was 9.6 versus 1.3 letters. The mean reduction in CST from the baseline was 153 μm versus 18 μm (*p* < 0.001), and elevated intraocular pressure occurred in 11.5% and 15.6% of the CLS-TA and control groups, respectively. Cataract AE rates were also similar (7.3% vs. 6.3%, respectively). The control group received sham injections. However, this leaves the basic question unanswered: Would administering triamcinolone periocularly in a different way or directly into the vitreous achieve better results? [28]

#### 3.1.5. Intravitreal Route

A high drug concentration can be obtained by an injection into the vitreous. Currently, intravitreal injections are the most frequently performed invasive ophthalmic procedure in the world due to the introduction of VEGF inhibitors. Although uveitis is not one of the most common indications for their use, they have often proved effective. Uveitis can lead to the development of choroidal neovascularization, so treatment with VEGF inhibitors is the treatment of choice, but anti-VEGF agents are often used successfully for inflammatory ME. Nevertheless, steroids are the primary medications administered intravitreally for uveitis. Unlike previous routes of drug administration, there is a possibility of infectious pathogens entering the vitreous, causing a risk of endophthalmitis. This risk is especially significant in the case of steroids (up to about 0.15–0.5% of injections) [29,30,31].

##### Steroids

Currently, some steroids are used intravitreally, differing primarily in their duration of action but sharing common side effects. Although the longest possible duration of action is always crucial for treating ME, treatment should not begin with the longest-acting steroids. The initial use of shorter-acting drugs prevents a surprise increase in IOP and the potential need for a vitrectomy to remove the steroid from the vitreous chamber. Despite the known differences between steroids, specific pharmacokinetics in particular patients have hardly been estimated, and there can be notable differences in the duration of action after using the same drug. Although triamcinolone is assumed to be shorter acting than Ozurdex^®^, it is impossible to predict the specific duration of action before its first use in a patient.

Triamcinolone acetonide is one of the most common intravitreally injected steroids that is used off-label, and there are many pharmacological triamcinolone products on the market. For many years, it was believed that the benzyl alcohol (preservative) content increased the risk of sterile endophthalmitis, especially in the presence of uveitis. Although this cannot be fully ruled out, studies have also suggested that the triamcinolone crystal size is more important than the preservative composition [32]. Due to limited solubility of triamcinolone, the desired concentration in the vitreous lasts for about three months after a single injection. The most common dose is 4 mg, but 2 mg is also used. Significant improvement in visual acuity has been observed in about 50% of patients. In most cases, repeated injections are needed to avoid ME relapses. Cataract progression relates to the number of injections, and 4–5 administrations almost always result in cataract formation. Increased IOP (in 20–45% of patients) is usually transient and easily managed with IOP-lowering medication.

The remaining drugs in this group are steroid devices, which, by slowly releasing the active substance (dexamethasone or fluocinolone), ensure many months or years of action. Because the implantation of such devices is more difficult than intravitreal injections, comparatively major surgical complications can be expected. In a retrospective analysis of 1241 dexamethasone implantations, 1.69% procedures led to complications, including displacement of the implant by corneal decompensation or hypotony, especially with preexisting risk factors (e.g., in post-PPV eyes) [33].

Ozurdex^®^ (a 700-µg dexamethasone intravitreal implant; AbbVie, Chicago, IL, USA) is a polymer-based, sustained-release corticosteroid formulation implanted into the vitreous, which has the shortest duration of action among the polymer drug group. The HURON study found that the proportion of eyes with a vitreous haze score of 0 at week 8 was 47% with a dexamethasone implant and 12% with a sham (*p* < 0.001)—a benefit that persisted through week 26. However, in terms of ME, a significant decrease in CST was seen at week 8, which did not persist until week 26 [34].

In the POINT six-month trial, periocular triamcinolone acetonide (PTA), intravitreal triamcinolone acetonide (ITA), and an intravitreal dexamethasone implant (IDI) were directly compared with randomization at 1:1:1. The study aimed to measure their influence on ME relative to the proportion of CST at baseline and CST at eight weeks (CST at 8 weeks/CST at BL) assessed with optical coherence tomography (OCT). Reductions of 23%, 39%, and 46% for PTA, ITA, and IDI were observed, respectively; thus, the intravitreal route was superior, but with no difference between drugs. The risk of an IOP of ≥24 mm Hg was higher in the intravitreal treatment groups than in the periocular group (95% CI: 1.83, 0.91–3.65 and 2.52, 1.29–4.91 for ITA and IDI, respectively), with no significant difference between the two intravitreal treatment groups [35].

Ozurdex^®^ insertion must be repeated more often than with Retisert^®^ or Iluvien^®^, so it is inconvenient to compare adverse events with those for other sustained-release devices. In a retinal vein occlusion (RVO) 24-month trial, cataract progression was observed in almost 40% of patients and an IOP increase in about 35% [36]. Since implantations can lead to vitreous infections, they can increase the risk of endophthalmitis compared to other steroid devices.

Iluvien^®^ (a 0.19 mg fluocinolone acetonide intravitreal implant; Alimera Sciences Ltd., England, UK) continuously releases 0.2 mg/day of fluocinolone to the posterior segment of the eye over a 36-month period. In a 36-month randomized trial, the time to first recurrence of uveitis was substantially longer than for the sham group—657 versus 70.5 days. ME was an additional outcome—more than two times fewer patients had investigator-determined ME in the treatment group than in the sham group (13.0% vs. 27.3%). It is worth noting that this result was not statistically significant (*p* = 0.079) and was mentioned only in the main text of the paper. Adjunctive treatment was received by 95.7% of patients in the sham group and 57.5% in the treated group, but the impact of these additional therapies on ME could not be determined. The lack of statistical significance could have been related to the size of the treated group—only about 60% of people had edema at the baseline, so the sample size was not sufficient for confirming ME. IOP-lowering medications were needed for 42.5% of the treated eyes and 33.3% of the sham group. Glaucoma surgery was needed for 11.9% of the treated eyes and 5.7% of the sham group. Furthermore, 73.8% versus 23.8% of phakic eyes required cataract surgery, respectively, in the treated and sham groups [37].

Retisert^®^ (a 0.59 mg fluocinolone acetonide intravitreal implant; Bausch & Lomb, Rochester, New York, NY, USA) is a sustained-release system designed to deliver corticosteroids inside the eye for up to 30 months. ME was a secondary efficacy outcome in a 36-month historically controlled clinical trial; since there was no control group, reduction of ME areas relative to ME in the fellow eyes were analyzed. Uveitis recurrence was reduced in implanted eyes from 62% (during the one-year preimplantation period) to 4%, 10%, and 20% (during the one-, two-, and three-year postimplantation periods, respectively). At the one- and three-year visits, the CME area was reduced in 86% and 73% of implantation cases compared with 28% and 28% in fellow nonimplanted eyes, respectively. An 80% reduction in systemic medications was needed as an adjunctive therapy during the postimplantation period. However, it was difficult to accurately trace the changes in concomitant treatment and their potential impact on ME from the article text. A greater than 10 mm Hg increase in IOP was noted in 67% of patients after implantation, and 40% of patients required glaucoma surgery. Over 90% of patients required cataract surgery in the fluocinolone implanted eyes during the study (compared to 20% in the fellow phakic eyes) [38].

##### VEGF Inhibitors

An increasing number of anti-VEGF agents have entered the market. Although a few have been registered for ophthalmology, this registration does not apply to uveitic ME; therefore, treatment, regardless of the choice of drug, is off-label. For patients who are steroid intolerant or phakic, VEGF inhibitors could be the treatment of choice. VEGF is a major vascular permeability factor and is heavily involved in the development of uveitic ME of various origins [39]. Given how frequently VEGF inhibitors are used in ophthalmology, it may come as a surprise that so little scientific evidence has been published for their efficacy in uveitic ME. Unlike steroid preparations, no VEGF inhibitors have such long durations of action. Currently, bevacizumab, ranibizumab, and aflibercept are used for other indications, but it is worth noting that anti-VEGF drugs—particularly brolucizumab–have the potential to cause inflammation, including occlusive retinal vasculitis [40]. No specific anti-VEGF treatment regimen for uveitic ME has been proposed so far. It is not known whether VEGF inhibitors should be used for people with infectious uveitis and ME, in cases where the use of immunosuppressing drugs may be harmful. It is also unclear whether it is necessary to wait for the inflammation to decrease or be eliminated before starting this treatment. Some studies have reported positive effects of anti-VEGF drugs (ranibizumab, bevacizumab, and aflibercept) on uveitic ME; the administration of subsequent injections was usually associated with a relapse or worsening of edema measured by OCT [41,42,43].

##### Immunomodulatory Agents

Methotrexate—an antifolate antimetabolite—can be injected into the vitreous to avoid systemic manifestations of adverse effects when inflammation is still active. In a prospective case study with 15 participants who all had intermediate or posterior ME or panuveitis, 400 µg/0.1 mL methotrexate was administered, and it improved ocular inflammation scores. On average, macular thickness decreased from 425 to 275 µm over the six-month period of observation. One third of patients relapsed at a median time of four months, but reinjection was as effective as after first injection. One (pseudophakic) patient developed corneal decompensation—which could be treated with topical folinic acid [44]. Although a larger study was conducted, it provided no detailed data regarding ME [45]. Cataracts are likely to develop less frequently with methotrexate than with steroid drugs, but due to short treatment times and small sample populations, it was impossible to identify the incidence of cataract as an adverse event in those groups.

Intravitreal sirolimus—an mTOR inhibitor—may be effective in some cases of uveitic ME; however, the SAVE-2 trial failed to achieve statistical significance in reducing ME. It may be crucial to select the right patients for sirolimus treatment of ME, since some participants showed significant improvement and others worsened during treatment [46].

### 3.2. Systemic Treatment

#### 3.2.1. Steroids

Systemic steroids are very effective for treating uveitic CME, but their use is limited due to their numerous systemic side effects. Discontinuation of medication often results in a recurrence of edema, necessitating re-treatment and associated adverse events. It is important to identify a personalized minimal effective dose—usually starting with 0.5–1 mg of prednisone/kg or an equivalent dose of other steroids [47]. The most used medications include oral prednisone and methylprednisolone. Systemic steroids are chosen more frequently for bilateral cases than for unilateral cases. ME is one of the factors contributing to the choice of oral steroids for uveitis, but oral corticosteroids are also one of the major strategies in relapse in uveitis without ME [48]. The side effects of systemic steroids are numerous and usually dose-dependent; therefore, chronic oral use is almost always destructive to the human body, from the skeletal system to the brain [47].

#### 3.2.2. Immunomodulatory Agents

Systemic treatment with immunosuppressive agents can effectively reduce ME associated with active inflammation. However, patients should be made aware that immunomodulating drugs are not panaceas for ocular complications. Evaluating immunosuppressive therapy in uveitis is very challenging. Only 19 randomized clinical trials of immunomodulating drugs for intermediate and posterior uveitis could be found in the medical databases, but these studies did not always present ME data, or the researchers observed positive trends toward reducing macular thickness but without statistical significance [49]. In the absence of relevant data, a retrospective analysis was warranted. This was conducted by the SITE study, which examined the past use of antimetabolites, T-cell inhibitors, alkylating agents, and other immunosuppressives based on the medical records of approximately 9250 uveitis patients at five tertiary centers over 30 years [50]. Of more than 1500 eyes, 52% showed improved visual acuity of at least the equivalent of two lines on an ETDRS chart [51]. Negative prognostic factors—snow banking (not snowballs), posterior synechiae, and hypotony—were also identified. The SITE study aimed to check whether the most-used immunosuppressive drugs led to increased cancer-related or overall mortality. The results suggested that tumor necrosis factor inhibitors could increase mortality, but this was not evident in patients treated with azathioprine, methotrexate, mycophenolate mofetil, ciclosporin, systemic corticosteroids, or dapsone (only cyclophosphamide was an exception among immunomodulatory agents) [52].

#### 3.2.3. Biologic Agents

No completed or ongoing clinical trials have confirmed or contradicted the efficacy of anti-TNF agents for uveitic ME [53]. However, based on some reports, it seems that, in at least some patient groups, subcutaneous TNF-alpha inhibitors may be effective for ME when used at standard doses [54].

#### 3.2.4. Interferons

In a two-pronged study, either interferon beta 44 mg was administered subcutaneously three times weekly or 20 mg MTX was administered subcutaneously once weekly; macular thickness decreased by a mean of 206 µm in the interferon group but increased by 47 µm in the methotrexate group (*p* < 0.0001) [55].

Interferon alpha2a has been used successfully to treat Behçet’s disease and other types of uveitis (about 60% efficacy in reducing inflammation). It can be even more effective for treating uveitic ME—one study reported control of ME in more than 80% of patients receiving subcutaneous interferon alpha2a [56]. Major but rare side effects (in about 5% of patients) include severe depression, neutropenia, and optic neuritis [56].

Currently, in many countries, access to the abovementioned interferons is very limited.

## 4. Surgical Treatment—Pars Plana Vitrectomy

Surgical treatment of uveitic ME remains a third-line therapy in most cases due to the significant risk of complications. However, pars plana vitrectomy should be considered for patients for whom the accumulation of inflammatory cytokines in the vitreous plays a dominant role. Some patients withstand vitrectomy surprisingly well, such as those with Fuch’s syndrome and severe vitritis. Heterogeneous visual improvement after uveitis treatment applies to almost all therapeutic modalities but is especially significant for PPV patients. It can be difficult or even impossible to distinguish the effects of haze and edema reduction. In many cases, the change in visual acuity may be attributed to a reduction in inflammation. In fact, post-PPV CST may remain near baseline with possible intravitreal drug pharmacokinetic deterioration; for example, triamcinolone acetonide has an 18.6-day half-life in non-vitrectomized eyes versus only 3.2 days in vitrectomized eyes [57]. A direct comparison of general treatment and PPV favored systemic medication [58].

Ophthalmologists should be aware that in some patients with uveitis, inflammation can increase as a result of surgery, usually requiring increased doses of previously used drugs or the initiation of more extensive therapy. The risk–benefit ratio is often low, so the decision to perform PPV (e.g., in patients with inflammatory epiretinal membranes) should not be made too hastily.

The effect of PPV on the pharmacokinetics of intravitreal drugs is not well understood and has been investigated mainly in animal studies. However, it seems that the duration of therapeutic drug action in vitrectomized eyes is quite short, meaning that, in some cases, VEGF inhibitors may be administered as often as every two weeks [59]. Thus, PPV potentially increases the role of long-acting drugs such as dexamethasone implants in this patient group.

## 5. What to Consider When Choosing a Treatment

Many factors should be considered when choosing a treatment, since the effectiveness of treatments may vary from patient to patient. When deciding on a specific treatment method, the questions in Table 2 can be used in conjunction with the short pros/cons listed in Table 3.

### 5.1. Bilateral Versus Unilateral CME

Avoidance of general side effects from chronic medication use favors the choice of local treatment, especially when the edema affects only one eye. When ME is bilateral, the benefits of local treatment are less obvious, but it can still prevent many side effects. An ophthalmologist’s decision not to treat both eyes simultaneously can also reduce the patient’s comfort, leading to more frequent visits. Although there is no conclusive evidence of an unfavorable response to bilateral treatment with intravitreal injections in both eyes, ophthalmologists should be aware that the incidence of complications in uveitis is higher than with routine administration of VEGF inhibitors for age-related macular degeneration (AMD) [60]; hence, the simultaneous administration of intravitreal steroids in both eyes may not be the optimal choice.

### 5.2. Age

Treatment at a young age is a negative prognostic factor, although uveitic ME is more common at an advanced age [61]. Young people not only develop a more aggressive form of the disease more frequently, but due to their longer potential life spans, they may experience more frequent relapses, and their retinas, despite morphological changes, will have to function for longer periods. The treatment for young people may therefore have to be very intensive. ME, unlike retinal neovascularization, does not cause rapid and permanent vision-threatening changes; however, especially for young patients, quick initiation of effective treatment is highly recommended.

### 5.3. Phakic Status

This factor relates to the previous one. Although ME is more critical than the potential for cataract formation, it is important to remember that the lens removal procedure itself may exacerbate inflammation. Moreover, removing a young person’s lens is a greater injury than it would be for a person with presbyopia. The use of multifocal intraocular lenses is not a good choice during cataract surgery in patients with uveitis, mainly for retinal reasons. Both for the patient and the ophthalmologist, because of the possible need to perform a vitrectomy, both epiretinal membrane (ERM) and internal limiting membrane (ILM) peeling are technically much more difficult in the presence of such lenses. 

Ocular complications of cataract surgery in uveitis can be significant, and in some groups of patients—especially younger patients—even the use of biological drugs does not significantly reduce this risk [62]; thus, intravitreal steroids should be avoided if the prevention of cataract surgery is desirable.

### 5.4. Economy

The choice of treatment does not always depend on strictly medical reasons. The scope of reimbursement for particular methods, or their availability in a given country, may key factors determining the method of treatment.

### 5.5. Combined Treatment

In the absence of available evidence-based medicine (EBM) data on combination therapies for uveitic ME, there are no clear indications in this regard. Available drug interaction data should be consulted to avoid interactions that may cause side effects or show similar mechanisms of action. One possible option is the use of a local corticosteroid along with systemic immunomodulatory and/or biological therapy.

## 6. Conclusions

It is impossible to identify a single treatment regimen that covers most clinical situations in uveitic ME, and regimens can be error-prone in many cases; therefore, the author recommends asking some basic questions for each patient in order to choose a course of action that is both safe and effective. The questions should primarily relate to the patient’s safety.

In many of the studies cited above, only half the patients responded well to ME treatment. Discussion with the patient is therefore important for deciding on an optimal treatment, including complex combined therapy, and this may take time and many attempts. If the patient is dissatisfied with the effects of a treatment and changes his doctor, previous attempts to identify an optimal treatment may be unnecessarily repeated. Cooperation based on trust is particularly important for treating ME, as it is for other chronic diseases. Acute treatment usually involves the use of steroids via various routes of administration, but long-term treatment should avoid steroid drugs because of their common side effects.

## Figures and Tables

**Table 1 jcm-10-04133-t001:** Common infectious causes of uveitis and ME (in alphabetical order) along with treatment options.

Pathogen	Examples of Treatment Regimens
*Bartonella* sp.	Treatment remains controversial. Doxycycline 100 mg bid, alone or in combination with rifampin 300 mg bid; fluoroquinolones; or macrolides + steroids (e.g., Prednisolone 60 mg/day). Treatment should continue for a few weeks [5].
*Borrelia* sp.	Oral doxycycline 100 mg bid or intravenous ceftriaxone 1 g/day + steroids (e.g., oral prednisolone 1 mg/kg/day) [6].
*Herpes* sp.	Oral Valacyclovir 1–3 g/day or acyclovir 5 × 800 mg/day + intravitreal foscarnet 2.4 mg/0.1 mL twice weekly [7].
*Mycobacterium tuberculosis*	Multidrug therapy with four drugs (isoniazid, rifampin, ethambutol, and pyrazinamide) according to the country’s health policy [8].
*Treponema pallidum* (syphilis)	Intravenous aqueous penicillin G 18–24 MU/day every 4 h for 10–14 days + oral or intravenous steroids [9].
*Toxocara* sp.	Poor visual outcomes are common despite treatment: albendazole + steroids or vitrectomy in severe cases [10].
*Toxoplasma* sp.	Oral six-week course of clindamycin, pyrimethamine “ sulfadiazine, or trimethoprim/sulfamethoxazole + a tapering course of oral prednisolone (1 mg/kg) [11].

**Table 2 jcm-10-04133-t002:** Questions facilitating the choice of treatment of uveitic ME.

Questions	Remarks
1. Has the patient been treated for ME? What were the effectiveness and complications of this treatment? Can the dose be adjusted to improve them? Is it better to repeat them, or does the response to this treatment suggest a need for change?	Previous treatment effects can be crucial when choosing an appropriate treatment regimen. Be sure to ask whether the patient has been treated previously by other ophthalmologists.
2. What complications should I especially avoid in this patient; for example, due to (a) advanced glaucoma changes, (b) age, (c) general diseases, (d) accompanying ocular changes, (e) the mental state of the patient, and/or (f) the expected compliance with medical recommendations?	ME is usually not the patient’s only problem. Chronic use of steroids or interferons in general can be dangerous in terms of the patient’s physical condition and mental health.
3. How often should I monitor and treat the patient—may I miss side effects or the need for additional treatment due to too infrequent follow-up visits?	IOP generally increases shortly after administration of periocular/intravitreal steroids; hence, after 1–2 weeks, it is worth checking the scale of this increase. VEGF inhibitors may not be effective for more than one month in many CME cases. This should be checked early in OCT, especially when starting therapy.
4. Is the patient a steroid responder for IOP?	If so, avoid local steroids unless pharmacological anti-glaucoma treatment is expected to be sufficient to normalize the IOP. In this case, do not start vitreous injections. Topical or posterior subtenon administration will be safer, although less effective, especially for the first option.
5. In the event of cataract formation, will it be safe to implant an artificial intraocular lens?	If not, try to avoid periocular/intravitreal steroids or intravitreal methotrexate. Administer oral steroids in carefully controlled doses.
6. Can a change in macular retinal morphology, especially a reduction in macular thickness, improve visual acuity?	Often, based on previously observed values, the patient’s prognosis of improved vision can be estimated. Sometimes aggressive treatment will only improve the OCT cross-section.
7. Do both eyes need ME treatment, or is there a case for focusing on treating the eye with the best prognosis or only the one with edema?	Treatment of both eyes can often be easier with systemic medication.
8. Do the treatment results so far suggest that a combined therapy may be needed?	Systemic and local medications can complement each other, but they reduce the comfort of therapy.
9. Is inflammation still active? Is intense inflammation the main cause of the edema?	A cytokine storm is not a good time to focus on the ME itself. If the inflammation is not under control, ME treatment may be ineffective or have only short-term effects.

**Table 3 jcm-10-04133-t003:** Advantages and disadvantages of individual therapeutic options for uveitic ME. Signs in the table: -, +/-, +, ++, +++ are ranked from least to most favorable, respectively.

Treatment	Efficacy	Safety	Supported by EBM Data	Duration of Action (Single Administration)	Cost
Topical steroids	-	++	-	-	+++
Topical NSAIDs	-	+++	-	-	+++
Subconjunctival triamcinolone	+	++	+	+	+++
Periocular/subtenon triamcinolone/methyloprednisolone	++	+	++	+	+++
Intravitreal triamcinolone	+++	+	++	++	+++
Ozurdex^®^	+++	+	++	++	+
Iluvien^®^	+++	+	++	+++	+
Retisert^®^	+++	+	++	+++	+
Intravitreal VEGF inhibitors	+	++	+	+	++
Systemic steroids	++	+	++	-	+++
Systemic immunomodulatory drugs	++	++	+	+/-	+++
Systemic biologic treatments	++	++	+/-	+/-	++

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
