# Peer review of "Update on the Management of Uveitic Macular Edema"

_jcm, 2021, doi:10.3390/jcm10184133_

Round 1
Reviewer 1 Report
In this review, author compiled the information on current options for treatment of uveitic macular edema (UME). In this review, author tried to focus on non infectious uveitis and covered various treatment options like local and systemic drugs, including steroids, immunomodulators, VEGF inhibitors and biological drugs. Here are few points that helps to improve the manuscript:
- The title of the review "How to manage macular edema in uveitis?" looks premature. Suggested title might be " An recent updates on management of uveitic macular edema".
- There are typo in manuscript for example in line 57 title of table mentioned as Table 2 however the title of table should be table 1 and In line no. 125 injection injection written two times.
Author Response
Thank you very much for your review.
The title has been changed.
Professional proofreading / editing was done to ensure higher quality English in the article.
Kindest regards,
Slawomir Teper
Reviewer 2 Report
This paper has relevant and well-organized content, but the poor English grammar makes it very difficult to read and review.
Author Response
Thank you very much for your review.
I am sorry for English grammar - professional proofreading / editing was done to ensure higher quality English in the article.
Kindest regards,
Slawomir Teper
Round 2
Reviewer 1 Report
Author responded to my query.